# Land Use and Climate Change Altered the Ecological Quality in the Luanhe River Basin

**DOI:** 10.3390/ijerph19137719

**Published:** 2022-06-23

**Authors:** Yongbin Zhang, Tanglei Song, Jihao Fan, Weidong Man, Mingyue Liu, Yongqiang Zhao, Hao Zheng, Yahui Liu, Chunyu Li, Jingru Song, Xiaowu Yang, Junmin Du

**Affiliations:** 1College of Mining Engineering, North China University of Science and Technology, Tangshan 063210, China; zyb063009@yeah.net (Y.Z.); songtanglei@stu.ncst.edu.cn (T.S.); zhenghao@stu.ncst.edu.cn (H.Z.); liuyh@stu.ncst.edu.cn (Y.L.); lichunyu@stu.ncst.edu.cn (C.L.); songjingru@stu.ncst.edu.cn (J.S.); yangxiaowu@stu.ncst.edu.cn (X.Y.); 2Hebei Tangshan High Resolution Earth Observation System Data and Application Center, Tangshan 063210, China; fanjihao@htwyysjhbyxgs.wecom.work (J.F.); dujunmin@htwyysjhbyxgs.wecom.work (J.D.); 3Aerospace Wanyuan Cloud Data Hebei Co., Ltd., Tangshan 063300, China; 4Tangshan Key Laboratory of Resources and Environmental Remote Sensing, Tangshan 063210, China; 5Hebei Industrial Technology Institute of Mine Ecological Remediation, Tangshan 063210, China; 6Hebei Key Laboratory of Mining Development and Security Technology, Tangshan 063210, China; 7Qinhuangdao City Surveying and Mapping Brigade, Qinhuangdao 066000, China

**Keywords:** ecological quality, time series, remote sensing ecological index, Google Earth Engine, climate change, land use

## Abstract

Monitoring and assessing ecological quality (EQ) can help to understand the status and dynamics of the local ecosystem. Moreover, land use and climate change increase uncertainty in the ecosystem. The Luanhe River Basin (LHRB) is critical to the ecological security of the Beijing–Tianjin–Hebei region. To support ecosystem protection in the LHRB, we evaluated the EQ from 2001 to 2020 based on the Remote Sensing Ecological Index (RSEI) with the Google Earth Engine (GEE). Then, we introduced the coefficient of variation, Theil–Sen analysis, and Mann–Kendall test to quantify the variation and trend of the EQ. The results showed that the EQ in LHRB was relatively good, with 61.08% of the basin rated as ‘good’ or ‘excellent’. The spatial distribution of EQ was low in the north and high in the middle, with strong improvement in the north and serious degradation in the south. The average EQ ranged from 0.58 to 0.64, showing a significant increasing trend. Furthermore, we found that the expansion of construction land has caused degradation of the EQ, whereas climate change likely improved the EQ in the upper and middle reaches of the LHRB. The results could help in understanding the state and trend of the eco-environment in the LHRB and support decision-making in land-use management and climate change.

## 1. Introduction

Natural ecosystems have been disturbed by rapid global climate change and modern industrialization. Climate change increases the frequency of extreme weather events [1,2], causing huge losses. Meanwhile, escalating human activities make climate change more severe and put more pressure on ecosystems and human well-being [3]. Climate change and human activities are changing ecosystem structure and functions [4,5] and making the eco-environment more vulnerable [6]. The expansion of the impervious surface area invades ecological land, which affects the original carbon cycle [7]. Greenhouse gasses from industrial production also aggravate global warming [8]. Deforestation and climate change have reduced resilience in the Amazon rainforest [9]. In the Qinghai–Tibet plateau, the expansion of built-up land contributed to the degradation of the ecological quality in some regions [10]. In the Yangtze River basin, climate change and ecological restoration projects improved the environmental condition [11]. A variety of signs show that climate change and human activities are affecting the eco-environment. Recently, the impacts of climate change and human activities has attracted more attention from the international community. In the 2030 Agenda for Sustainable Development, the United Nations adopted 17 Sustainable Development Goals (SDGs). Most of them are closely related with ecological quality. For instance, SDG15 is aimed at protecting terrestrial biodiversity. As ecological quality affects sustainable development and human well-being, the assessment of ecological quality is a key step for us to decide how to protect it.

Previous research has established excellent methods to evaluate ecological quality. These include single indexes such as the normalized difference vegetation index (NDVI) [12] and fractional vegetation coverage (FVC) [13] that are usually used to assess the condition of vegetation, and synthetic models like the eco-environmental quality (EEQ) [14] and the Ecological Index (EI) [15], which have shown great potential in assessing ecological environmental quality precisely and comprehensively. However, there are still some limitations. EEQ uses an analytic hierarchy process (AHP) for weight determination, which is easily affected by subjective factors. EI is unable to obtain a fine-scaled spatial distribution as the statistical data might be at an administrative division scale [16]. In addition, it is difficult to collect large amounts of data to evaluate the Ecological Footprint of a large-scale natural ecosystem for a long period of time [17]. The Remote Sensing Ecological Index (RSEI) is an ingenious method for ecological quality assessment. It is fully based on remote sensing data, which are convenient to obtain. It uses principal component analysis (PCA) instead of AHP for weight regulation, which is determined by the data rather than by a subjective factor. It also provides a result with high spatial and temporal resolution [18,19]. Since RSEI was developed, it has been applied in ecological quality assessment of cities [20], basins [21], nature reserves [22], and various regions around the world. Some research even improves the model to make it more suitable for specific regions [23]. Generally, applications proved the feasibility and efficiency of RSEI.

Current research usually constructs RSEI with a few Landsat images at a few year intervals around specific dates [24,25,26]. The interval may cause the changes in some years to be ignored. Moreover, the differences–including but not limited to the different sensors on the Landsat satellites (Landsat 5, 7, and 8, etc.)–could reduce the consistency of the observation [27,28]. The Moderate-resolution Imaging Spectroradiometer (MODIS) provides a stable and long-term dataset, which has a higher temporal resolution (2-day revisit period) than Landsat. In addition, cloud cover has a major influence on the accuracy of surface reflectance, especially in the growing season [29]. The Google Earth Engine (GEE) is a cloud-based platform that integrates enormous remote sensing datasets and has a superior image processing capability [30]. GEE greatly reduces the hardware requirements for remote sensing computation. As a result, we can use the cloud-free function and mean value composition in GEE to reduce some error [31].

The Luanhe River is part of the Haihe River Basin and the largest river in Hebei province, China. The LHRB ecosystem plays a vital role in northern China. It is one of the major water sources as well as an ecological barrier for the Beijing–Tianjin–Hebei area [32]. However, climate change and human activity have been increasingly severe in the recent decades, bringing about more ecological and environmental issues [33,34]. The long span in latitude makes LHRB cross different climatic zones with great variant climate conditions [35], therefore, climate could be one of the main factors affecting ecological quality. Related research usually applied RSEI on a smaller scale [19,36] without such characteristics. In addition, current studies rarely discuss the impact of climate change intensively [37,38], and the eco-environment monitoring of LHRB is limited to single indicators [9,39,40]; thus, a long-term and comprehensive eco-environment evaluation is lacking. Our study provided a comprehensive ecological quality assessment based on remote sensing ecological index from 2001 to 2020 with GEE and discussed the impacts of land use and climate change, which could help to better understand the eco-environment of the LHRB and support decision-making about land use management and climate change. To support ecology protection and help SDGs’ achievement in the LHRB, this research aims to: (1) monitor and quantify the ecological quality condition by RSEI, (2) investigate the changing trend of the ecological quality, (3) analyze how land use and climate change affect the ecological quality. First, we constructed time-series RSEI from 2001 to 2020 on GEE to quantify the condition and investigate the spatial distribution of ecological quality in the LHRB. Then, we introduced the coefficient of variation, Theil–Sen trend analysis, and Mann–Kendall test to evaluate the variation degree and changing trend of ecological quality. Finally, we discussed the feasibility of the RSEI model and the response of ecological quality to land use and climate change.

## 2. Materials and Methods

### 2.1. Study Area

The Luanhe River Basin is between 115.2~119.4° E and 39.2~43.4° N and crosses the junction of semi-humid and semi-arid regions [41], covering an area of more than 45,000 km^2^ that takes up nearly 20% of Hebei province.

The length of the river is approximately 888 km [32]. It crosses the Inner Mongolia Plateau, the mountainous area of North China, and the North China Plain–flowing through 27 cities and counties [33]–and has rich mineral resources [42,43]. The climate in the northwest is temperate continental, while that in the southeast is temperate monsoon, with low temperature and rare precipitation in winter and elevated temperature and sufficient precipitation in summer. The annual precipitation and average temperature are approximately 488.4 mm and 7.0 °C [32]. The current main land use types are grasslands (45.29%), forestlands (25.93%), cultivated lands (23.60%), construction lands (3.07%), water bodies (0.69%), bare lands (0.64%), shrublands (0.45%), and wetlands (0.33%). The elevation ranges from 0 to 2220 m.

According to local records, the LHRB is divided into three parts: the upper reaches (UR) above Zhangbaiwan, the lower reaches (LR) below Luanzhou, and the middle reaches (MR) between them (Figure 1).

### 2.2. Data

#### 2.2.1. MODIS Data

The NDVI was extracted from the 16-day 1 km Vegetation Index product (MOD13A2 V6, DOI: 10.5067/MODIS/MOD13A2.006), which is corrected for atmospheric conditions and chooses the best available pixel value in the 16-day period [44]. The land surface temperature (LST) was extracted from the 8-day 1 km L3 LST product (MOD11A2 V6, DOI: 10.5067/MODIS/MOD11A2.006), which is the average pixel value of daily LST in the 8-day period [45]. The tasseled cap wetness (WET) and normalized difference built-up and soil index (NDBSI) were calculated based on the 8-day 500 m Land Surface Reflectance product (MOD09A1 V6, DOI: 10.5067/MODIS/MOD09A1.006), which selects the best value from an 8-day composition based on cloud cover, view angle, etc. [46].

To avoid the influence of cloud cover, we used the cloud-masking function to composite cloud-free images [47]. Then, to ensure vigorous plant growth and avoid the influence of the uncertainties of time on some indicators, we filtered all the images over the growing season (from June to September) of every year from 2001 to 2020 and used mean value composition to construct the yearly continuous time-series data [48]. The NDVI and LST were resampled with surface reflectance to the same 500 m resolution.

#### 2.2.2. Other Data

The land-use data in 2000, 2010, and 2020 were from GlobeLand30, a free and open 30-m resolution land-use dataset developed by China, and includes 10 land cover classes. The dataset is based on multispectral images; the total accuracy of GlobeLand30 2020 is 85.72% and provided by the online portal of GlobeLand30 of the National Geomatics Center of China (DOI: 10.11769).

The monthly precipitation and temperature from 2001 to 2020 were extracted from the 1-km monthly precipitation and mean temperature datasets for China (1901–2020) [49,50], which were evaluated to be dependable by 496 national weather stations across China and were provided by the National Tibetan Plateau Data Center (http://data.tpdc.ac.cn, accessed on 26 January 2022). Mean annual precipitation (MAP) and mean annual temperature (MAT) were calculated based on the monthly precipitation and temperature.

The 90-m Shuttle Radar Topography Mission (SRTM) DEM was acquired from Geospatial Data Cloud (https://www.gscloud.cn/, accessed on 1 April 2021) to extract the boundary of LHRB. We divided the LHRB into three parts according to local records, terrain, and the boundary of sub-basins.

### 2.3. Quantification of Ecological Quality

Remote sensing provides an efficient and convenient way to monitor the ecosystem. The RSEI can comprehensively quantify ecological quality based on the Pressure–State–Response (P–S–R) model. The *RSEI* contains four indices—greenness, wetness, dryness, and heat–which are represented by the *NDVI*, *WET*, *NDBSI*, and *LST*, respectively, and uses the principal component analysis to regulate the weights. Using the cloud platform GEE, *RSEI* can be quickly constructed.

*NDVI* is a common index for monitoring vegetation health. The *NDVI* and the day *LST* from MOD13A2 V6 and MOD11A2 V6 products were selected to represent the greenness and heat in the *RSEI*. The *NDVI* and surface reflectance were converted into real values with a scaling factor of 0.0001. The *LST* was first converted into Kelvin degrees with a scaling factor of 0.02 and then converted into Celsius degrees by subtracting 273.15 [19]. *WET* indicates the moisture of soil and vegetation and can be calculated as follows [51]:(1)WET=c1⋅B1+c2⋅B2+c3⋅B3+c4⋅B4+c5⋅B5+c6⋅B6+c7⋅B7
where *B*1–7 indicates the surface reflectance of the top seven bands of MOD09A1 V6 product and *c*1–7 indicates the tasseled cap transformation coefficient [51]; *NDBSI* is the mean value of soil index (*SI*) and index-based built-up index (*IBI*), which represents dryness with the brightness of bare soil and buildings, respectively. The equations for calculating *NDBSI* are as follows [52]:(2)$$BI=\frac{\left(B6+B1\right)-\left(B2+B3\right)}{\left(B6+B1\right)+\left(B2+B3\right)}$$
(3)$$IBI=\frac{\frac{2\cdot B6}{B6+B2}-\frac{B2}{B2+B1}+\frac{B4}{B4+B6}}{\frac{2\cdot B6}{B6+B2}+\frac{B2}{B2+B1}+\frac{B4}{B4+B6}}$$
(4)$$NDBSI=\frac{BI+IBI}{2}$$

A water mask is required during the computation as *RSEI* is not suitable for regions covered by a large area of water. Therefore, the modified normalized difference water index (*MNDWI*) was used to create the water mask, which can be calculated with MODIS surface reflectance images using the equation below [53]: (5)$$MNDWI=\frac{B7-B4}{B7+B4}$$

While the units or dimensions of each index are different, there must be a normalization for all the indexes before *RSEI* is constructed. After normalization, a principal component analysis (PCA) was utilized to determine the weights of each index by the data’s characteristics instead of the researcher’s subjective perspective. Finally, the *RSEI* was constructed by normalizing the first principal component (PC1):(6)$$RSEI_{0}=\mathrm{PCA}1(f[NDVI,LST,WET,NDBSI])$$
(7)$$RSEI=\frac{RSEI_{0}-RSEI_{\min}}{RSEI_{\max}-RSEI_{\min}}$$
where *RSEI*_0_ is the first principal component of the *NDVI*, *LST*, *WET*, and *NDBSI*. *RSEI*_min_ and *RSEI*_max_ are the minimum and maximum values of *RSEI*_0_.

The value of RSEI ranges from 0 to 1; the higher the value, the better the ecological quality. To evaluate the ecological quality more efficiently, the RSEI is divided into five levels based on previous research [19]: poor (0–0.2), fair (0.2–0.4), moderate (0.4–0.6), good (0.6–0.8), and excellent (0.8–1).

### 2.4. Coefficient of Variation

Coefficient of variation (CV) can measure the variation degree of time series data, which reflects the time-dependent variation of spatial data, and assesses the stability of time series [54]. CV can be calculated using the equation below [54]:(8)$$\mathrm{CV}=\frac{\sigma }{\mu }$$
where *σ* is the standard deviation and *μ* is the average EQ value of every single pixel.

A higher CV indicates the EQ in this region fluctuated more dramatically than in other regions during the period. According to previous research, when CV > 0.20, we consider the time-series EQ to be at “High variation”, followed by “Relatively high variation” (0.15–0.20), “Medium variation” (0.10–0.15), “Relatively low variation” (0.05–0.10), and “Low variation” (0.00–0.05) [55].

### 2.5. Theil–Sen Median Trend Analysis and Mann–Kendall Test

Theil–Sen Median trend analysis (Sen’s slope) is a robust trend estimator for long time series, which is a non-parametric statistic method that is not sensitive to outliers [56]. Therefore, we used Sen’s slope to calculate the trend of EQ. The statistic that Sen’s slope provides is *β*, which is the median value of the slopes in different periods. *β* can be calculated as follows [57]:(9)$$\beta =\mathrm{median}\left(\frac{x_{j}-x_{i}}{j-i}\right),\forall j>i$$
where *x_j_* and *x_i_* represent the EQ value of each pixel in year *j* and year *i*. When *β* > 0, there is an increasing trend. When *β* < 0, there is a decreasing trend.

Sen’s slope is usually combined with the Mann–Kendall (M–K) test and is popular in vegetation study [58,59], meteorology [60], etc. While Sen’s slope is for trend analysis, Mann–Kendall is used to assess the significance of the trend. M–K test is also a non-parametric method that can resist influence from outliers and doesn’t require the data to obey any certain distribution [61]. M–K test defines the *Z* value, which can be calculated as follows [62,63]:(10)$$Z=\left\{\begin{array}{ll} \frac{S}{\sqrt{\mathrm{Var}(S)}} & \left(S>0\right)\\ 0 & \left(S=0\right)\\ \frac{S+1}{\sqrt{\mathrm{Var}(S)}} & \left(S<0\right) \end{array}\right.$$
(11)$$S=\sum_{i=1}^{n-1}\sum_{j=i+1}^{n}\;sign\left(x_{j}-x_{i}\right)$$
(12)$$sign(\theta )=\left\{\begin{array}{cc} 1 & \left(\theta >0\right)\\ 0 & \left(\theta =0\right)\\ -1 & \left(\theta <0\right) \end{array}\right.$$
(13)$$\mathrm{Var}(S)=\frac{n(n-1)(2n+5)}{18}$$
where *x_j_* and *x_i_* represent the EQ value of each pixel in year *j* and year *i*, and *n* is the length of the time series, which is 20 in this research. *S* is an intermediate variable. When *n* ≥ 0, *Z* approximately obeys normal distribution. Given a significance level of 95%, if |*Z*| > 1.96, we consider the trend to be significant, else it is not significant. Moreover, the M–K test can also express the trend. A positive *Z* means an increasing trend and a negative *Z* means a decreasing trend.

## 3. Results

### 3.1. Spatiotemporal Distribution of Ecological Quality in LHRB

The areas with lower EQ were mainly located in the north of LHRB, which is part of the Inner Mongolia plateau, where an arid climate and sparse vegetation could lead to lower EQ. In the middle and lower reaches around the urban and mining area of Chengde and Qianan, the EQ was also low. In contrast, the regions with higher EQ were distributed in the south of the upper reaches and in the north of the middle reaches, which belong to the mountainous area of North China and are covered by more forestlands. In general, the EQ was low in the north and high in the middle. The average EQ of the whole basin ranged from 0.58 to 0.64 (Figure 2 and Figure 3), with a mean value of 0.60 over the last 20 years, indicating the ecological quality in the LHRB was relatively good.

In addition, there appeared to be a trend toward improvement in EQ, as indicated by the time series (Figure 3) and spatial distribution (Figure 2). In the upper reaches, the average EQ kept increasing from 0.49 to 0.59, with fluctuations between 2006 and 2010 and a low peak in 2009, which was similar to the characteristics of the whole LHRB. The increasing trend of LHRB and middle reaches is probably due to the wetter climate and the restoration of vegetation. In the middle reaches, the average EQ rose from 0.71 to 0.72, with an increasing trend from 2001 to 2013 and a decreasing trend from 2013 to 2020, which stayed relatively stable. In the lower reaches, the average EQ declined from 0.55 to 0.53. The average EQ of lower reaches reached the highest value (0.58) in 2007 and decreased to a low peak (0.49) in 2014, then steadily increased after 2014. These fluctuations probably resulted from climate change, and the urban expansion and mining activity could be the reason for degradation.

### 3.2. Ecological Quality Assessment in LHRB

The average EQ of LHRB was assessed as ‘good’. The proportions of various levels ranging from ‘poor’ to ‘excellent’ were 2.13%, 15.38%, 21.62%, 50.95%, and 9.92%, respectively. The average EQ of the upper, middle, and lower reaches were rated as ‘moderate’, ‘good’, and ‘moderate’, respectively. In the upper reaches, the proportions were 3.65%, 25.88%, 25.17%, 37.70%, and 7.60% in sequence. In the middle reaches, there were no pixels rated as ‘poor’. The percentages of other levels ranging from ‘fair’ to ‘excellent’ were, in order, 0.47%, 13.90%, 71.88%, and 13.41%. In the lower reaches, the proportions varying from ‘fair’ to ‘good’ accounted for 3.96%, 82.63%, and 13.41%, respectively (Figure 4). This result shows that the EQ in the middle reaches was much better than that in the upper and lower reaches, while the EQ in the top of the upper reaches was much worse than that in other regions, which is probably related to different land use and climate conditions.

Figure 5 reveals that in LHRB, there has been an 80.49%, 33.62%, and 5.42% decline in the ‘poor’, ‘fair’, and ‘good’ areas, respectively, and a 28.74% and 93.09% rise in ‘moderate’ and ‘excellent’, respectively. In the upper reaches, the ‘poor’ and ‘fair’ areas exhibited a downward trend. On the other hand, the ‘moderate’, ‘good’, and ‘excellent’ areas have increased (Figure 5b). There are very few ‘poor’ and ‘fair’ areas in the middle reaches, with a slight upward trend. There has been an increase in ‘moderate’ and ‘excellent’ areas, but a decrease in ‘good’ (Figure 5c). In lower reaches, the areas with ‘poor’ and ‘excellent’ EQ are very small and have decreased, as with the ‘fair’ and ‘good’ areas. Nonetheless, there has been an increase in the ‘moderate’ areas (Figure 5d). The change in area proportions of different levels seemed to be stable, with sharper changes in the lower reaches. The ‘good’ area presented a downward trend till 2014 and an upward trend after 2014, with fluctuations in 2007 and 2012, having the same characteristics as the average EQ. Climate change and human activity probably caused these changes. These results suggest that the EQ has improved in the upper reaches, has degraded in the lower reaches, and was relatively stable in the middle reaches. The fluctuations also indicated that the EQ in the upper and lower reaches could be sensitive and vulnerable to external factors such as land use and climate change.

We generated a Sankey diagram (Figure 6) to illustrate how the different levels changed over these years. From 2001 to 2020, the transitions from lower to higher levels covered an area of 15,569.94 km^2^, whereas the reverse transitions covered 2512.72 km^2^, showing obvious transitions of lower levels outward and higher levels inward (Figure 6a). 2094.62 km^2^ of ‘poor’ were turned to ‘fair’, accounting for 69.55% of the total area of the ‘poor’ level in 2001. The conversion of 4623.88 km^2^ of ‘fair’ to ‘moderate’ accounted for 62.91% of the former. 52.86% of ‘moderate’ (4168.80 km^2^) were transformed into ‘good’. 18.53% of ‘good’ (4332.25 km^2^) were converted to ‘excellent’.

In the upper reaches, the transition from lower levels to higher levels and higher levels to lower levels comprised 12,827.93 km^2^ and 482.98 km^2^, accounting for 49.68% and 1.87% of the total area, respectively. In the middle reaches, the ratio was 13.43% (2473.67 km^2^) and 9.84% (1812.04 km^2^). In the lower reaches, the ratio was 15.17% (119.94 km^2^) and 23.00% (181.82 km^2^). 

In general, the LHRB, as well as the upper and middle reaches, had more areas that transitioned to higher levels, whereas the lower reaches had more areas that converted to lower levels. The transitions also indicated that the EQ has improved in the upper and middle reaches, while it degraded in lower reaches.

### 3.3. Coefficient of Variation

CV describes the variation degree of the time-series EQ. The spatial distribution was high in the north and low in the center (Figure 7), which is similar to the opposite distribution pattern of EQ (Figure 2). The average CV of LHRB was 0.09, with an 8.92% area of high variation. The average CV in the upper, middle, and lower reaches was 0.12, 0.05, and 0.09, respectively. The area of high variation accounted for 14.95% in the upper reaches, whereas in the middle and lower reaches, high variation only accounted for 0.16% and 4.56%, respectively (Table 1). The EQ of LHRB was relatively stable, with only 8.92% in high variation, notably in the upper reaches. This indicates that the EQ in the upper reaches has fluctuated more significantly than other regions and the ecosystem in the upper reaches is more sensitive. The main land use type in the upper reaches is grassland, which is sensitive to external influence. We consider that climate change and human activity probably made the grasslands undergo degradation and restoration, which resulted in fluctuating EQ.

### 3.4. Trend Analysis of Ecological Quality

To further determine the time-dependent trend of ecological quality, the Mann–Kendall (M–K) test was analyzed (Table 2). The *Z* value confirmed our hypothesis that the EQ of the upper and middle reaches, as well as the entire basin, increased over the 20-year period. However, the EQ of lower reaches decreased.

Theil–Sen trend analysis and Mann–Kendall test were introduced (Sen + M–K test) to assess the trend and dynamic of ecological quality variation in LHRB (Figure 8b), Table 3). The Sen + M–K test results indicate that area of improvement was much larger than the area of degradation. The area of strong improvement accounted for 42.52% of the basin, while serious degradation made up 7.35%. The majority of LHRB showed signs of EQ improvement, particularly in the upper reaches. Regions with EQ degradation were mainly found in the south of the middle and lower reaches, especially among the urban areas (Figure 8c). Area of strong improvement accounted for 52.67%, 29.21%, and 11.28% in the upper, middle, and lower reaches, respectively. The EQ in the lower reaches deteriorated considerably, with serious degradation accounting for 38.82%. In conclusion, EQ has improved dramatically over the last 20 years.

## 4. Discussion

### 4.1. Feasibility of Ecological Quality Assessment with RSEI

In the LHRB, current research for eco-environment monitoring is limited to vegetation index [39], water quality [64], meteorology [40], hydrology [9], and other single indicators; our study provided a comprehensive ecological quality evaluation based on RSEI. In addition, previous studies usually construct RSEI using an interval of a few years [24,26], which cannot detect the variation of years between the multiphase RSEI. In this study, we carried out a 20-year long-term monitoring by constructing RSEI every year.

The principal of RSEI is actually based on the P–S–R model. NDBSI represents the pressure from human activity. NDVI represents the state of the eco-environment. LST and WET represent the climate in response of environment [65]. The loading of four indexes on PC1 shows that NDVI (greenness) and WET (wetness) have positive effects on ecological quality, whereas LST (heat) and NDBSI (dryness) have negative effects on ecological quality (Table 4), which is consistent with the principal of the RSEI and other studies [18,19,66]. The eigenvalue contribution of PC1 ranged from 82.35% to 92.85% (Table 4), indicating that PC1 contains most of the information of the four indexes and PCA is appropriate for weight determination. Moreover, the contribution of PC1 is even larger than many related studies [21,67].

The construction of RSEI using the Google Earth Engine is efficient. Therefore, we consider that the model is feasible for ecological quality assessment of LHRB and other regions, even at a larger scale.

### 4.2. Response of Ecological Quality to Land Use and Climate Change

#### 4.2.1. Response of EQ to Land Use

(1)EQ of various land use types

Water body and wetlands were excluded from the comparison in view that the water could affect the EQ value. As GlobeLand 30 only contains data from 2000, 2010, and 2020, we used the land use of 2000 to substitute that of 2001.

In 2001, the average EQ of different land use types ranged from 0.26 to 0.72. The land use type with the highest EQ here was forestlands (0.72 ± 0.12), followed by shrublands (0.60 ± 0.19), cultivated lands (0.54 ± 0.18), grasslands (0.54 ± 0.23), construction lands (0.54 ± 0.13), and bare lands (0.27 ± 0.22). In 2010, EQ of forestlands, shrublands, grasslands, and cultivated lands increased, while that of construction lands and bare lands decreased. In 2020, EQ of all land use types showed improvement (Figure 9). 

The spatial distribution of land use is quite similar to that of EQ (Figure 2 and Figure 10). Forestlands is primarily in the center of the LHRB, while grasslands are throughout the upper and middle reaches. Cultivated lands are mostly distributed in the northwest and south. Construction lands are concentrated in the south, and the majority of bare lands are located in the north. The similar spatial distribution indicates the EQ and land use are likely related.

The spatial distribution of land-use is quite similar to that of EQ. Forestland has great ecological effects and provides a variety of ecosystem services including wildlife habitat, carbon storage, climate control, etc. [68]. As a result, forestland has the highest EQ. Cultivated lands and grasslands also have considerable ecological effects, although not as much as forestlands [69], even if their similar spectral features during the growing season may result in a similar average EQ. Moreover, the EQ of construction lands is also close to that of grasslands and cultivated lands. According to the land use in the LHRB, grasslands and cultivated lands are the main land use types in the north. Previous studies reported that drought would lead to plant trait losses [70], and drought conditions are severe in the north LHRB [71]. Thus, it is possible that the drought in the north LHRB causes the EQ of grasslands and cultivated lands to be as low as that of construction lands (Figure 9). Additionally, the bare lands are mostly located in the upper reaches, which makes the EQ of bare lands even poorer than that of construction lands. When examining satellite imagery, we found some open-pit mines in the upper and middle reaches in 2020 (Figure 8b). The damage that mining brings to the environment is extreme [72,73]. The metal and mining industry plays an important role in the local economy, with hundreds of mining companies in Chengde and Qianan [42,43], putting huge pressure on the local ecology.

Since there is a substantial growth of construction lands (Figure 10) and a significant downward trend of EQ in such areas (Figure 8b,c), we speculate that the expansion of construction lands should lead to EQ degradation.

(2)EQ variation and the land use change

We extracted the regions where the EQ was seriously degraded and strongly improved (Figure 8) and then superimposed these regions with the transitions of various land use types. In the regions where the EQ strongly improved, the proportion of vegetation cover reaches approximately 99%. The land use exhibited little variation.

From 2001 to 2010, the most significant transition was from cultivated lands to forestlands and grasslands. From 2010 to 2020, grasslands converted into cultivated lands. From 2001 to 2020, the area of forestlands was stable, while that of cultivated lands and grasslands changed from 4865.36 km^2^ and 8604.79 km^2^ to 4498.94 km^2^ and 8821.42 km^2^, respectively. The most noticeable changes are the transitions between cultivated lands, forestlands, and grasslands. These regions are mostly located in the semi-arid area where the grasslands are relatively more sensitive. According to Figure 9, the EQ of grasslands has seen more improvement than that of cultivated lands. Therefore, we consider the transition from cultivated lands to grasslands likely improved the EQ.

In regions with serious degradation, vegetation still accounts for the largest proportion, but has dropped dramatically by 14.09%. The most obvious change is the conversion from cultivated lands to construction lands. From 2001 to 2020, the area of construction lands rose from 218.52 km^2^ to 520.50 km^2^, with 89.58% converted from cultivated lands and most of the conversion happening during the period from 2010 to 2020. Another significant change is the larger proportion of bare lands converted from grasslands in these regions. That is, the expansion of construction lands and bare lands and the reduction of cultivated lands and grasslands should be the main reason EQ degraded.

The average CV of various land-use types was in the sequence of bare lands (0.21) > grasslands (0.11) > cultivated lands (0.10) > construction lands (0.10) > shrublands (0.09) > forestlands (0.05). The lowest CV indicated the EQ of forestlands was relatively stable. Forestlands in LHRB have increased from 11,477.4 km^2^ to 11,815.7 km^2^, although the vegetation area has reduced. Therefore, we consider the forestlands to be important for the stability of EQ, and forestation probably mitigates the pressure from urban expansion.

Other studies found that land use could explain most of the spatial heterogeneity of the RSEI [67], and the expansion of construction land or impervious surface is the main reason that RSEI decreased [25,74], which is consistent with our result. Previous research also pointed out that urban expansion is a threat to habitats [75]. In this research, we found large amounts of cultivated lands converted into construction lands (Figure 11) in the lower reaches, which indicates the increasing human activity probably resulted in the EQ degradation in these areas. In the middle and lower reaches of LHRB, mining activities were frequent over the past 20 years. Some of the mining areas even experienced both mineral development and ecological restoration, with the EQ first decreasing and then increasing [76]. These mining activities also have great impact on the EQ.

Land-use change played a major role in the runoff of LHRB [9]. It has threatened the ecosystem in the LHRB and will keep affecting the ecosystem service in the future if there is no conservation strategy [32]. Our suggestion is to reduce the invasion of ecological and agricultural land and accelerate ecological restoration of the mining area.

#### 4.2.2. Response of EQ to Climate Change

From 2001 to 2020, the MAP ranged from 408.17 mm to 613.42 mm. Although it decreased dramatically in some years, the overall trend is increasing (Figure 12a,b). The MAT changed more rapidly than MAP, while the decreasing trend from 2007 to 2012 and the increasing trend from 2012 to 2017 are still obvious. Through the M–K test, the *Z* values show that the MAP is increasing (Figure 12b), indicating the climate in the LHRB is getting wetter. However, the MAT presents a non-significant trend. It seems that the MAT in most parts of the upper reaches is slightly decreasing, while in the middle and lower reaches it is likely getting warmer (Figure 12d). Comparing the change of MAP and MAT with that of EQ (Figure 3), it seems that their characteristics in variation are quite similar. The EQ and MAP of LHRB shared consistent turning points (such as their low peak in 2009 and turning point in 2002, 2016, and 2019, etc.), while MAT had opposite shifts around the same years. These changes indicated the fluctuation of the EQ is probably due to climate change, especially precipitation.

We calculated the Spearman correlation between EQ and MAT and that between the EQ and MAP by pixel to investigate the relationship between climate change and EQ. The results of the Spearman correlation analysis showed an increasing monotonic relation between the MAP and EQ in most parts of the LHRB (Figure 13), which is similar to the spatial distribution of the highly improved areas (Figure 8a). The monotonic relation indicated that the increasing precipitation could improve the EQ in most regions, while too much precipitation could have a negative effect on the EQ in the lower part of the middle reaches. As for the areas around construction lands and cultivated lands with more human activities, the impact of precipitation would be greatly reduced. The change of temperature probably had less influence on EQ, as the MAT had a significant negative correlation with EQ only in a few regions in the lower-middle part and a non-significant positive or negative correlation in most of the other regions.

In the whole LHRB, the land use did not show a dramatically significant change, while the overall EQ was improving with a monotonic relation between climate factors, thus, we think the climate change may affect the EQ more. The changing trend of climate factors is generally consistent with other studies [77].

In some areas, NDVI is usually the main factor affecting RSEI [19,66]. In this research, according to the loadings on PC1, LST and WET had more contribution to EQ than NDVI and NDBSI (Table 4), indicating that climate factors probably had more influence on the EQ in LHRB. Although LST was selected to characterize the climate response of the environment, the relationship between LST and MAT is complicated as there are too many factors that could affect it including different land-use types [78]. Previous research found the correlation between plant growth and temperature in northern China is not significant during the growing season [79,80], which means the MAT change likely cannot affect NDVI too much. As a result, the impact of MAT is limited.

WET has been used as a proxy of soil and vegetation moisture and can be easily influenced by precipitation [81,82]. Sufficient precipitation provides moisture for soil and vegetation, which would improve plant growth and affect NDVI [83]. Sometimes, too much precipitation could inhibit it because insufficient sunshine would inhibit photosynthesis [84]. The vegetation in the upper reaches of LHRB showed an improvement over the past few years according to recent research; this is correlated with precipitation, which grassland is more sensitive to [85]. With increased soil moisture, LST would decrease due to the water–heat balance [86]. On the other hand, decreased MAP would lead to EQ degradation. Thus, MAP can affect the EQ to a great extent.

The impacts of land use and climate change in LHRB have led to various results for different ecosystem functions. The frequent change in precipitation caused recurrent droughts and runoff reduction [87,88], while improving vegetation net primary productivity in the upper reaches [85]. Climate change and land-use change also reduced water yield and purification of LHRB [34]. In our research, from the perspective of RSEI, the overall EQ of LHRB is improved due to climate change.

Forestation can regulate the impacts of climate change and human activity through the carbon sink [89,90,91]. According to land-use change, the forestlands in LHRB have increased from 11,477.4 km^2^ to 11,815.7 km^2^ during the past 20 years, which illustrates the local government’s efforts. Because the ecological quality in the LHRB is sensitive to climate change, we suggest carrying out more forestation projects to enhance resistance against sharp fluctuations of climate change and mitigate the impacts of urban expansion.

In fact, sensitivity to climate change is also a limitation of RSEI. In this research, although we constructed RSEI with cloud-free and mean value composite images, the sharp changes in climate factors made the RSEI fluctuate abnormally for some years, which added uncertainty to the evaluation and should be improved in the future to enhance robustness. The land-use data only includes 2000, 2010, and 2020, which also adds uncertainty about land-use change and limits our purpose of long-term and continuous monitoring in this research. In addition, a fusion with Landsat datasets would improve the spatial resolution and accuracy.

## 5. Conclusions

This research constructed time-series RSEI from 2001 to 2020 to assess the EQ of LHRB and introduced CV and Sen + M–K to investigate the variation trend of EQ. Ultimately, the response of EQ to land use and climate change was discussed.

The results showed a strong spatial heterogeneity in the EQ of LHRB. The EQ was low in the upper reaches due to drought climate conditions, while it was improved by climate change. In the middle reaches, the EQ was higher and more stable, even though there were frequent mining activities during the period. In the lower reaches, EQ was low and degraded because the construction land kept expanding rapidly. The overall EQ was relatively good; it was altered by land use and climate change. The area of different levels kept converting over the past 20 years.

Through time-series analysis, we found climate change greatly affected the EQ of LHRB. The increasing precipitation improved the EQ in most regions, and the fluctuations in precipitation and temperature made the EQ fluctuate at the same time. Meanwhile, the expansion of construction land and warmer climate in urban areas led to the degradation. The EQ was sensitive to climate change and land-use change. Stopping the invasion of ecological and agricultural land could help to mitigate these impacts in the future.

## Figures and Tables

**Figure 1 ijerph-19-07719-f001:**
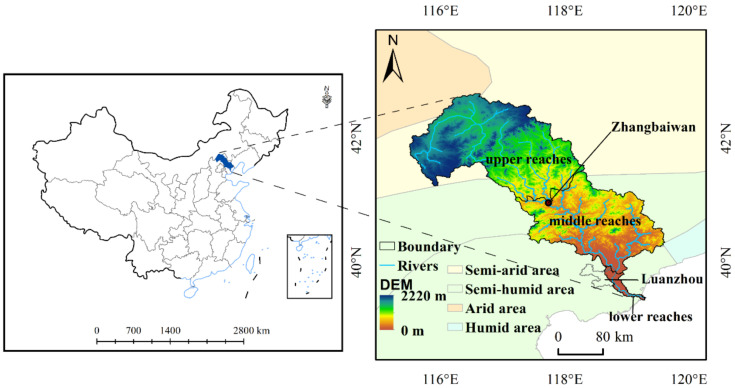
Geographic location of the study area.

**Figure 2 ijerph-19-07719-f002:**
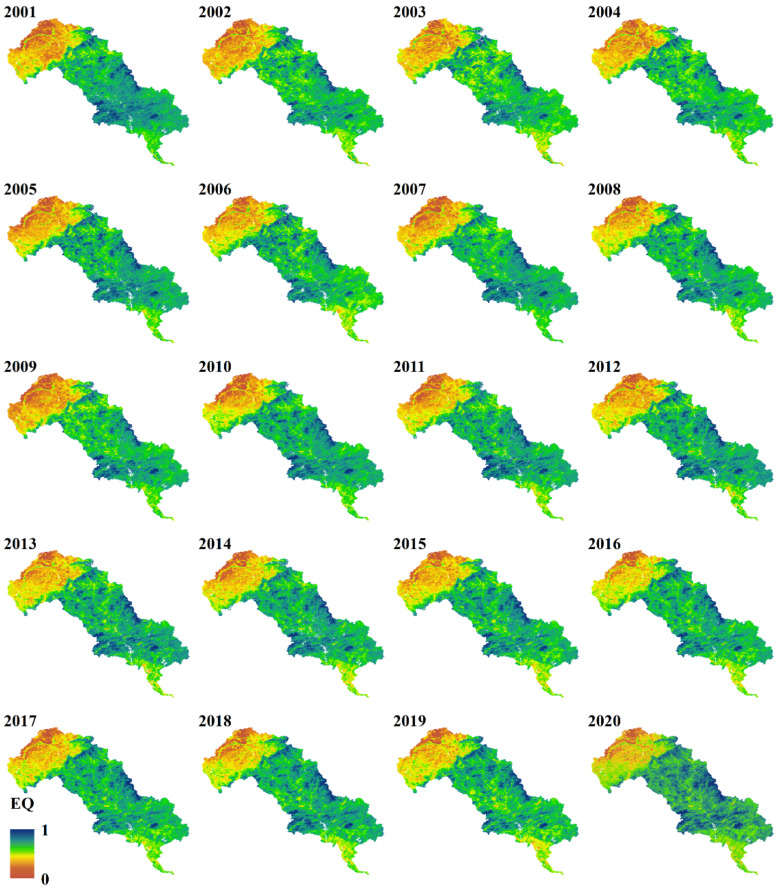
Spatial distribution of ecological quality of LHRB from 2001 to 2020.

**Figure 3 ijerph-19-07719-f003:**
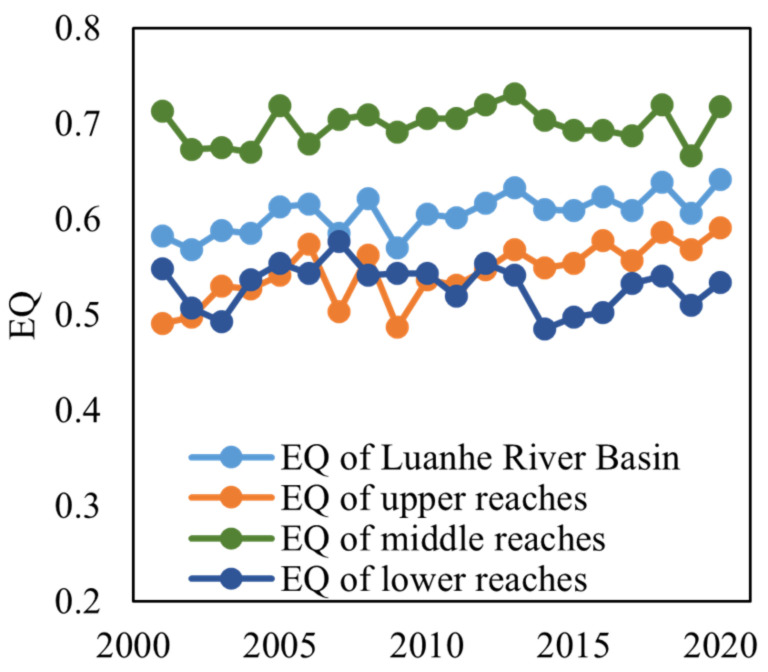
Time-series mean EQ of LHRB from 2001 to 2020.

**Figure 4 ijerph-19-07719-f004:**
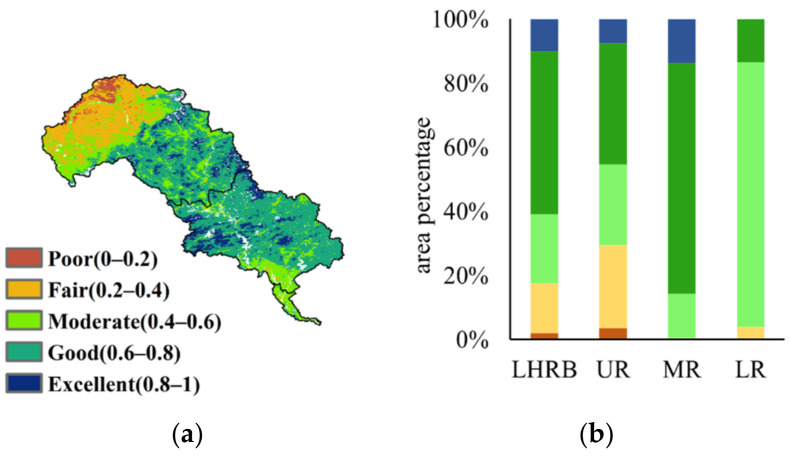
Ecological quality assessment of LHRB (**a**) spatial distribution, (**b**) area ratio bar chart.

**Figure 5 ijerph-19-07719-f005:**
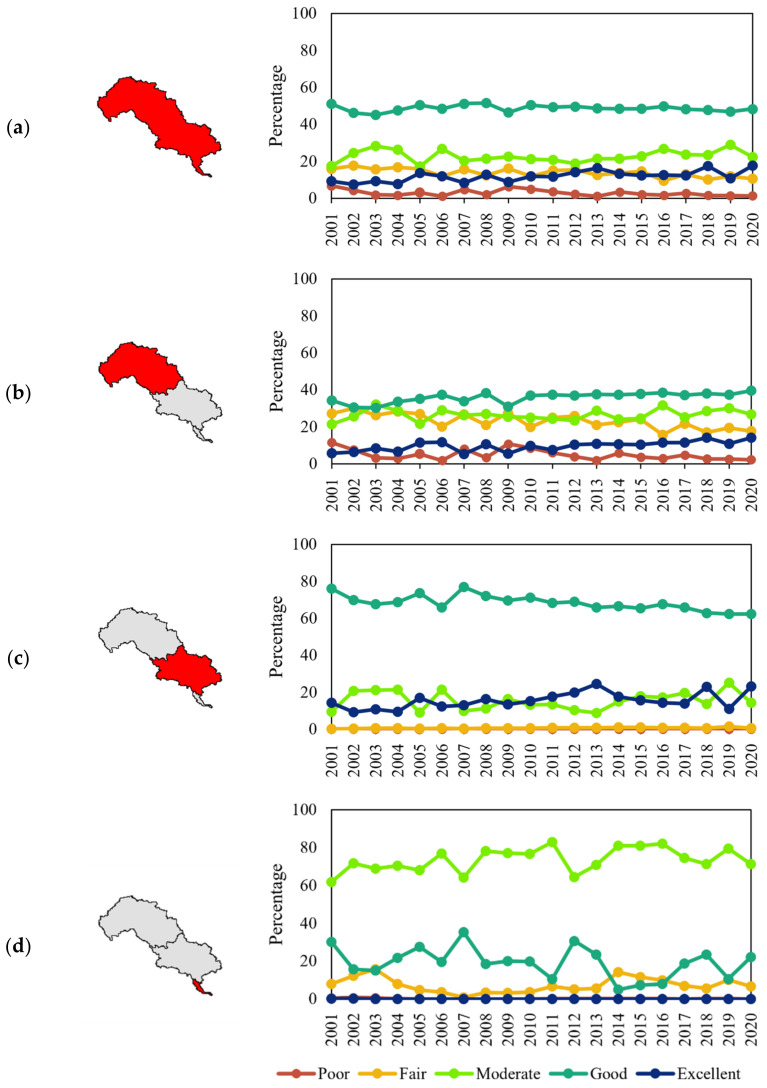
Proportions of different levels in time series (**a**) LHRB, (**b**) upper reaches, (**c**) middle reaches, (**d**) lower reaches.

**Figure 6 ijerph-19-07719-f006:**
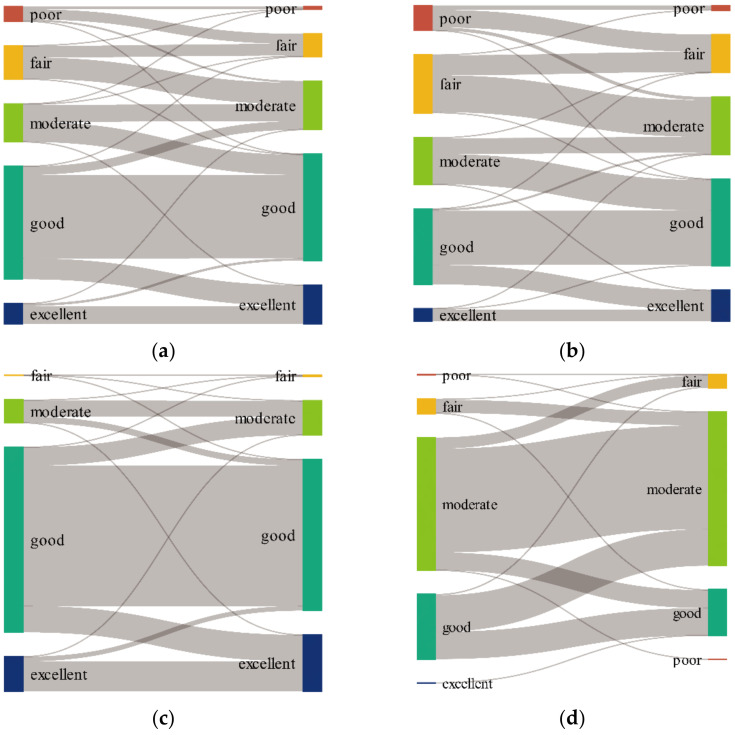
Area transitions (km^2^) of various levels from 2001 to 2020 in (**a**) LHRB, (**b**) upper reaches, (**c**) middle reaches, (**d**) lower reaches.

**Figure 7 ijerph-19-07719-f007:**
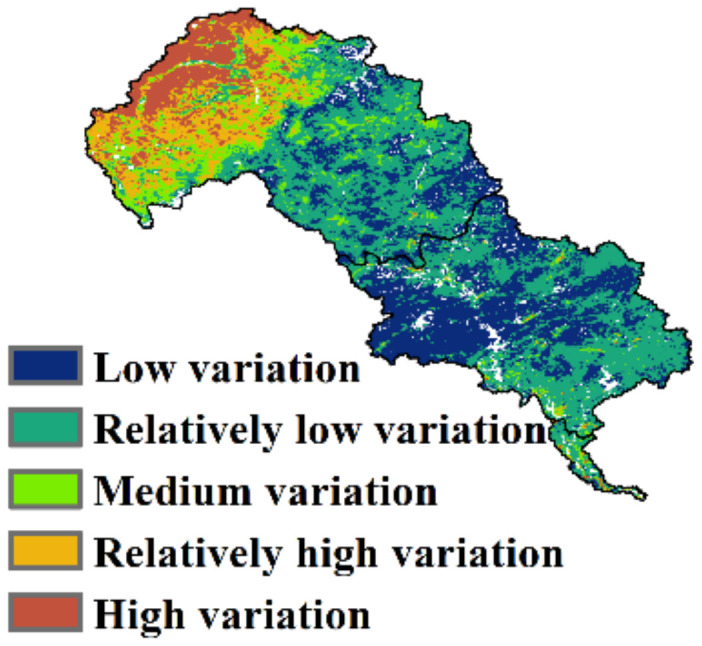
Spatial distribution of CV of EQ from 2001 to 2020.

**Figure 8 ijerph-19-07719-f008:**
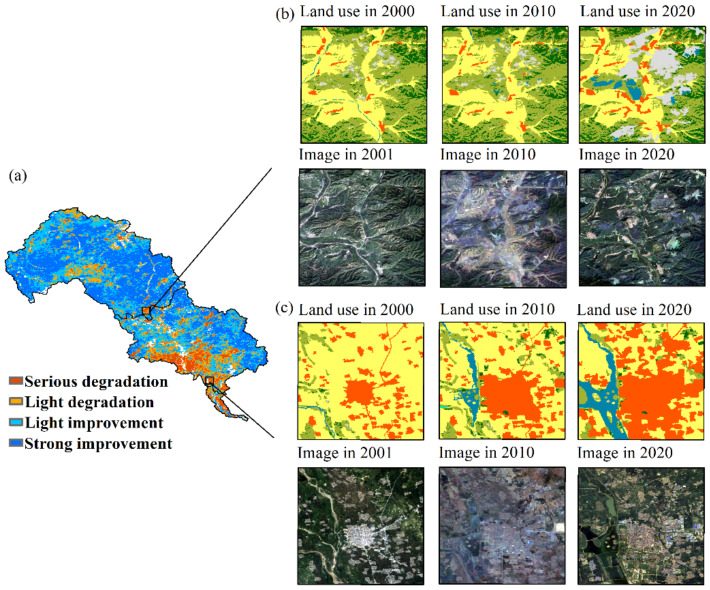
Trends of EQ change from 2001 to 2020 (**a**) result of trend analysis, (**b**) imagery comparison of a mining area in Chengde, (**c**) imagery comparison of an urban area in Qianan.

**Figure 9 ijerph-19-07719-f009:**
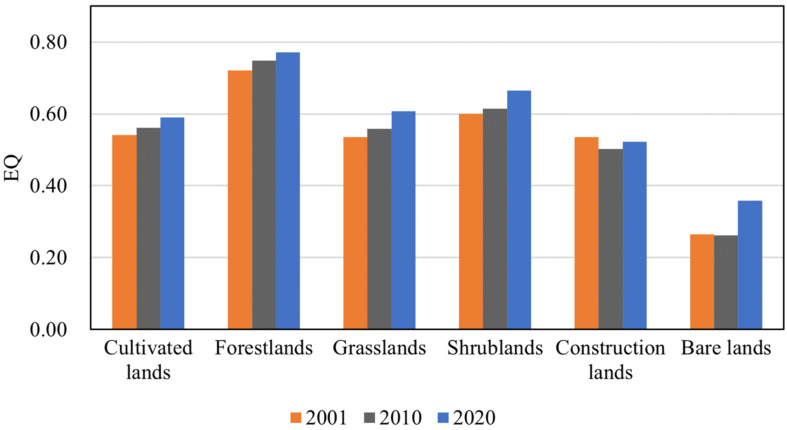
EQ of various land use types in 2001, 2010, and 2020.

**Figure 10 ijerph-19-07719-f010:**
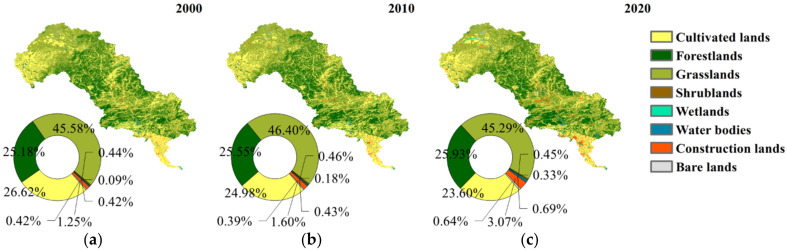
Land use in (**a**) 2000, (**b**) 2010, (**c**) 2020.

**Figure 11 ijerph-19-07719-f011:**
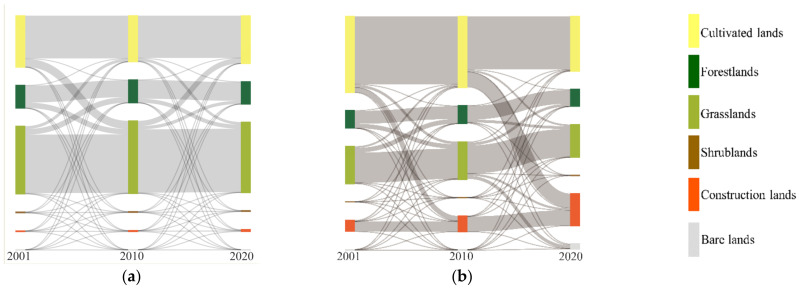
(**a**) Land use change in the regions with strong improvement; (**b**) Land use change in the regions with serious degradation.

**Figure 12 ijerph-19-07719-f012:**
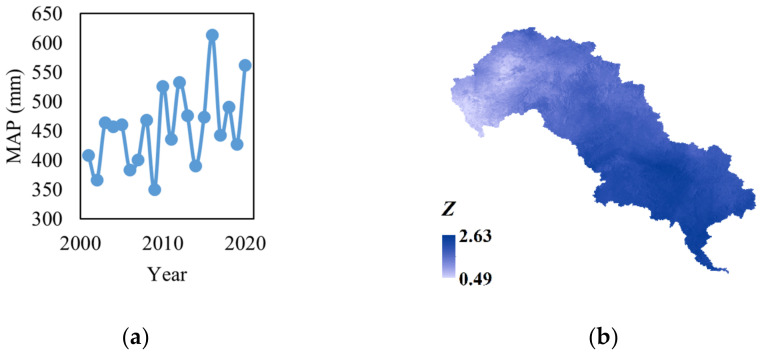
Trend of climate factors (**a**) MAP, (**b**) M–K test of MAP, (**c**) MAT, (**d**) M–K test of MAT.

**Figure 13 ijerph-19-07719-f013:**
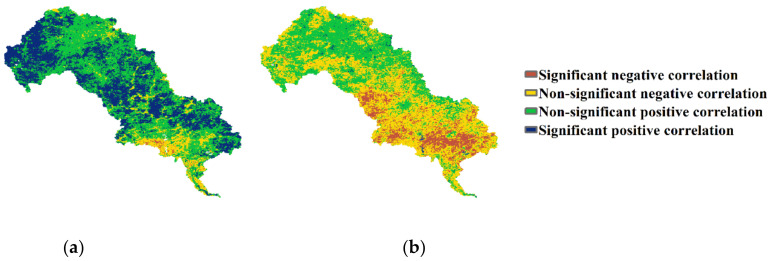
(**a**) Spearman coefficient between MAP and EQ from 2001 to 2020 by pixel, (**b**) Spearman coefficient between MAT and EQ from 2001 to 2020 by pixel (0.05 significance level).

**Table 1 ijerph-19-07719-t001:** Variation degree of EQ in different regions from 2001 to 2020.

CV	Variation Degree	Percentage			
		LHRB	UR	MR	LR
≤0.05	Low variation	29.32%	16.66%	45.60%	10.02%
0.05–0.10	Relatively low variation	42.80%	38.15%	48.76%	64.84%
0.10–0.15	Medium variation	10.10%	15.48%	2.05%	15.77%
0.15–0.20	Relatively high variation	8.85%	14.76%	0.43%	4.82%
>0.20	High variation	8.92%	14.95%	0.16%	4.56%

**Table 2 ijerph-19-07719-t002:** Results of Mann–Kendall test of EQ in different regions from 2001 to 2020.

Test	Regions			
	LHRB	UR	MR	LR
*Z*	2.89	3.70	0.72	−1.24

**Table 3 ijerph-19-07719-t003:** Trends of EQ change in different regions from 2001 to 2020.

*β* (Theil-Sen)	*Z* (Mann-Kendall)	Trend	Percentage			
			LHRB	UR	MR	LR
*β* < 0	*Z* < −1.96	Serious degradation	7.35%	1.98%	13.77%	38.82%
*β* < 0	−1.96 < *Z* < 1.96	Light degradation	15.74%	11.31%	21.49%	30.50%
*β* > 0	−1.96 < *Z* < 1.96	Light improvement	34.39%	34.05%	35.53%	19.40%
*β* > 0	*Z* > 1.96	Strong improvement	42.52%	52.67%	29.21%	11.28%

**Table 4 ijerph-19-07719-t004:** Results of principal component analysis.

Year	Contribution (%)		Loading on PC1			
	PC1	PC2	NDVI	LST	WET	NDBSI
2001	91.86	4.90	0.50	−0.61	0.52	−0.44
2002	89.98	6.49	0.46	−0.50	0.66	−0.43
2003	82.35	13.77	0.39	−0.42	0.60	−0.35
2004	85.74	10.49	0.46	−0.63	0.67	−0.41
2005	89.79	7.02	0.41	−0.53	0.56	−0.42
2006	82.36	13.45	0.39	−0.49	0.59	−0.36
2007	92.51	4.23	0.50	−0.61	0.59	−0.46
2008	88.15	8.23	0.42	−0.64	0.63	−0.44
2009	90.80	5.99	0.51	−0.53	0.60	−0.45
2010	92.85	3.92	0.43	−0.64	0.59	−0.45
2011	91.36	5.26	0.44	−0.59	0.62	−0.44
2012	90.31	5.99	0.42	−0.61	0.67	−0.46
2013	83.24	12.90	0.34	−0.35	0.48	−0.33
2014	87.67	8.94	0.42	−0.52	0.53	−0.40
2015	84.04	12.01	0.32	−0.45	0.52	−0.33
2016	86.75	9.20	0.40	−0.53	0.54	−0.36
2017	88.02	8.07	0.41	−0.54	0.57	−0.39
2018	86.81	8.88	0.39	−0.53	0.59	−0.38
2019	85.77	10.12	0.38	−0.55	0.62	−0.37
2020	82.71	13.52	0.32	−0.36	0.48	−0.33

## Data Availability

Not applicable.

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
