# Peer review of "Land Use and Climate Change Altered the Ecological Quality in the Luanhe River Basin"

_ijerph, 2022, doi:10.3390/ijerph19137719_

Round 1
Reviewer 1 Report
The authors provided a technique to monitor and assess ecological quality. The topic is interesting and meaningful. However, I think the quality of the presentation needs much improvement.
(1) Introduction does not clearly state the purpose of this study. In addition, although the previous studies were reviewed in Introduction, it is not clear how this study has novelty in comparison with them. Other authors did not select continuous data perhaps because of the low resolution.
(2) Satellite images taken between 2000 and 2020 are used to look at the time-series changes. The MODIS images selected by the authors are not of higher resolution and cloud coverage is not clear. The manuscript should indicate the data processing process. The change of climate and ecological environment is sometimes abnormal, and it can be considered the maximum effect of some indicators when monitoring the change of ecological environment.
(3) The whole Results section from 3.1 to 3.3 simply shows the results, which can actually be seen in a graph or table, and readers need to know some of the reasons for these results. Please explain and supplement the results in detail.
(4) From 2000 to 2020, the land use in the LHRB has undergoing great changes. However, the authors only compared the changes between 2000 and 2020 in the discussion part, without considering the fluctuations and causes during the period, which failed to achieve the purpose of long-term monitoring.
(5) These indices used in the manuscript are closely related to the presence and amount of precipitation and temperature changes. The research result of the references 32 and 33 cited by the authors are the relationship between climate and NDVI/LUCC in the growing season, and the indicators selected in this paper included LST. It is necessary to check how did this result come about and using weather data to correct the comparison.
I believe that the paper could be improved by incorporating the following recommendations.
(1) There are several inconsistencies in the tables. For example, there are two MR in Tables 1, 2 and 3.
(2) After that, try to position the paper against the past research so as to justify its added value and its originality. The authors have to be able to demonstrate how the paper differs from the past research. The fact that it is implemented in a different geographic context is not enough.
(3) Try to elaborate more on the conclusions. After your empirical analysis you should be able to generalize somehow your results. Try to open the discussion which now focuses explicitly at the study area without any implications for other areas. At the end you should make your paper appealing to readers around the globe.
Author Response
We appreciate your comments. We have read your comments carefully and revised the manuscript according to your comments point to point. Please see the attachment.

Reviewer 2 Report
It is very important to study the effects of land use and climate change on ecological environment. But this manuscript only used a simple assessment method to evaluate the ecological equality and did not analyze how affect the land use and climate change on ecological equatlity. The authors lack scientific understanding of this issue.
The details of the issues are as follows:
1. There is a lack of research review on the impact of climate change and human activities on the ecological environment in Section 1.
2. The introduction on the methods and models of ecological environment assessment is also very insufficient in Section 1.
3. Line92-96: these sentences should be the submission requirements and should not appear in the manuscript.
4. The description of Section 2.1 is not related to the research content, and there is no introduction of the climate and meteorological conditions, hydrological regime, land use and ecological environment of the study area.
5. Proposed to replace "Satellite data" by "MODIS data", because this manuscript does not employ any Satellite data other than MODIS data. In addition, Section 2.2.1 does not introduce the useful information (data period,spatio-temporal resolution etc.) of MODIS data on land use and vegetation.
6. Line 118: when the abbreviations first appear, please give its full name.
7. Line 136: what is the abbreviations "WET"?
8. Line 146: what is the abbreviations "BI"?
9. What is the meaning of "RSEI0" in Equation (6)?
10. What is the meaning of "PCA1" in Equation (6)?
11. Fig.4(b) has not axises.
12. What is the physical meaning of the CV for describing EQ?
13. Table 3 has not the bottom line.
14. The caption of Secition 4.1 is "Feasibility of ecological quality assessment with RSEI", but the contents of this section between lines 279 and 285 are not the description of the feasibility of EQ assessment with RSEI. In addition, line 283 "which is consistent with general cognitive and our expectation", this subjective understanding cannot be used as the criterion to judge the feasibility of this assessment index.
15. Section 4.2 should classed to Section 3 "Results", becasue these are the calculated results on understanding EQ in LHRB.
16. Comparing land use in 2000 and that in 2020, the land use between 2000 and 2020 is not almost change, What are the factors that affect the EQ better in LHRB?
17. This manuscript only used the indexes of the precipitation and temperature to characterize the climate change,which is very inappropriate.
18. The significant correlation between EQ and the indexes of precipitation and temperature does not mean that the indexes are the influencing factors of EQ change.
Author Response

(The authors gave the same response as above.)

Reviewer 3 Report
refer my attachment

Author Response

(The authors gave the same response as above.)

Round 2
Reviewer 1 Report
This revised manuscript has solved most of the questions. It can be published after careful revision of language and logical expression.
Reviewer 2 Report
The authors have done their best to revise the manuscript. I have no major comments and suggestions.
Reviewer 3 Report
i am satisfied with the corrections that have been made by the researchers. therefore, I do not hesitate to suggest that this article be taken to the next level.